# What triggers online help-seeking retransmission during the COVID-19 period? Empirical evidence from Chinese social media

Chen Luo[1,2], Yuru Li[3]*, Anfan Chen[4]*, Yulong Tang[5]

1 School of Journalism and Communication, Tsinghua University, Beijing, China, 2 Department of Communication, University of California, Davis, Davis, California, United States of America, 3 Centre for Media, Communication & Information Research, University of Bremen, Bremen, Germany, 4 School of Humanity and Social Science, University of Science and Technology of China, Hefei, Anhui, China, 5 Institute of Communication Studies, Communication University of China, Beijing, China

* yuru.li@foxmail.com (YL); caftsinghuaedu@gmail.com (AC)

**Data Availability Statement:** Data are available via figshare: https://doi.org/10.6084/m9.figshare. 12743726.v1 Statistical commands are also

## Abstract

The past nine months witnessed COVID-19's fast-spreading at the global level. Limited by medical resources shortage and uneven facilities distribution, online help-seeking becomes an essential approach to cope with public health emergencies for many ordinaries. This study explores the driving forces behind the retransmission of online help-seeking posts. We built an analytical framework that emphasized content characteristics, including information completeness, proximity, support seeking type, disease severity, and emotion of help-seeking messages. A quantitative content analysis was conducted with a probability sample consisting of 727 posts. The results illustrate the importance of individual information completeness, high proximity, instrumental support seeking. This study also demonstrates slight inconformity with the severity principle but stresses the power of anger in help-seeking messages dissemination. As one of the first online help-seeking diffusion analyses in the COVID-19 period, our research provides a reference for constructing compelling and effective help-seeking posts during a particular period. It also reveals further possibilities for harnessing social media's power to promote reciprocal and cooperative actions as a response to this deepening global concern.

## Introduction

According to the World Health Organization (WHO), COVID-19 has resulted in more than 14 million confirmed cases and nearly 0.6 million deaths worldwide by July 20th, 2020 [1]. Due to the destructive power and highly infectious feature of this new coronavirus, many countries have taken various kinds of measures to prevent virus transmission. Meanwhile, people use social media to acquire and exchange multiple types of information at a historic and unprecedented scale [2]. So far, some researches have concentrated on the effects of social media in this particular period. For example, harness the social media posts to predict infected case counts and inform timely responses under the infoveillance or infodemiology framework [3,4]; analyze the help-seeking posts to identify the characteristics of COVID-19 patients [5];

available via figshare: https://doi.org/10.6084/m9.figshare.12743750.v1

**Funding:** This study is supported by the Science Popularization and Risk Communication of Transgenic Biotechnologies Project (Grant code: 2016ZX08015002) to CL and AC, China Scholarship (Grant code: 201906210115) to CL, the 25th department funding of USTC (Grant code: DA2110251001) to AC, 2019 New Humanities Funding of USTC (Grant code: YD2110002015) to AC. The funders had no role in the study design, data collection, analysis, decision to publish, or manuscript preparation.

**Competing interests:** The authors have declared that no competing interests exist.

discuss the online censorship on social media from the risk communication perspective [6]. Among all the research, online help-seeking, as a practical approach to handle difficulties during public health emergencies, has a unique research significance. Firstly, when facing scarce medical resources during the initial outbreak stage, many suspected and confirmed COVID-19 patients were failing to be admitted to a hospital, many of them turned to social media fork help (e.g., sickbeds, testing kits, emotional comforts) and share information instantly [5]. Secondly, online help-seeking helps relevant actors to ease anxiety, build mutual aid networks, and make the most of the collective power to cope with risks [7,8]. Social support or so-called supportive communication in computer-mediated communication contexts has always been a hotly discussed issue [9]. COVID-19, as an epidemic affecting the whole world, provides a valuable chance to explore the mechanisms and effects of online help-seeking. Its practical and theoretical implications not only offer an insight into understanding help-seeking and promote problem-solving during a public health crisis but also enrich the strategies on how to develop compelling online help-seeking posts.

As Van Bavel et al. [10] suggested, social and behavioral science can be a beneficial framework to support the COVID-19 pandemic response. Online help-seeking and help-providing are typical reactions toward pandemic, encourage social connection to buffer the negative feelings of social isolation in the shelter-in-place period [10]. This study aims to explore help-seeking information diffusion on Chinese social media. As Wang et al. suggest, probing into information diffusion is the basis for understanding and managing the dynamics of content on social media [11]. Generally speaking, retweeting help-seeking posts can be interpreted as a kind of prosocial behavior, which means retweeters intended to help others by joining the retransmission process [12,13]. Meanwhile, online retweeting behavior is driven by a dual-process [14]. It is not only impelled by logical cues (e.g., deliberation based on facts) but also affected by intuitive cues (e.g., emotions) [10,14]. Previous studies provided limited insights on evaluating help-seeking information diffusion during health emergencies. Many studies followed the "sender-content" framework [15] but failed to develop a comprehensive content measurement for health crisis communication. To the best of our knowledge, this study is the first online help-seeking diffusion analysis during the COVID-19 period. We intended to combine logical cues and intuitive cues to replenish content measurement, makes it more plausible and suitable to the current situation. Our findings will provide a reference for social media help-seeking in a similar period in the future.

## Literature review and research hypotheses

### Retransmission of help-seeking posts and content characteristics

With the rapid development of social media, new possibilities of seeking help online keep emerging. More and more people are turning to cyberspace to seek specific social support in various domains like coping with daily hassles and battling a life-threatening illness [16]. Social network sites like Twitter [17], Facebook [18], Instagram [19] or online forums [20–22] increase the opportunities of receiving and providing social support from all sides. Retransmission, or the so-called retweeting, has always been seen as a crucial indicator of information diffusion in social media platforms [23]. Some studies treated retransmission as a judgment of the effectiveness of communication [11,24]. The retweeting behavior has been demonstrated as a conversational practice on social media platform [25]; a way to seek personal benefits from the social network [26]; a kind of prosocial behavior aims to offer help or provide advice to others driven by altruistic and reciprocal motivations [13]. Retransmission is critical in online help-seeking, especially in the epidemic context for two reasons. Firstly, retweeting is a typical one-to-many communication describing the degree of viral reach on social media [24].

More retransmission means more users receiving the message, thus increasing the chance of getting help. Secondly, from a practical point of view, facing the medical resources shortage and uneven distribution of medical facilities, widely disseminated help-seeking posts played the role of social warning, assisted official institutions in understanding urgent affairs as well as allocating supplies more effectively.

What are the driving factors behind online help-seeking messages retransmission? Most existing studies categorized driving forces into two dimensions: the sender factors and the content factors [11,23,27]. However, significant driving forces are varied by context. In this study, ordinary users are actively adopting the Weibo platform to express their needs and compete for public attention. Other types of users, including celebrities and organizations, are rarely asking for help online. In short, the main component of online help seekers is ordinaries, and their online identities are nearly the same. Thus, it would be more reasonable and significant to explore further in the content factor. There is plenty of research providing conceptual references, such as the depth of self-disclosure [16,28], different types of support messages [16,29], physical and emotional proximity to the target [30], the social capital stock of the help seeker [22]. Enlightened by existing experience, we will summarize the impelling factors of help-seeking information diffusion into five dimensions: completeness, proximity, support typology, disease severity, emotion, and elaborate them in the following sections. Under the dual-process perspective, the former four determinants stay close with the slow and deliberative process while the emotion is akin to the fast and intuitive cognitive process. In other words, retweeting behavior is propelled by "relying on reason" and "relying on emotion" simultaneously [14,31].

## Completeness

Completeness is bound up with credibility. Credibility has always been seen as perceived quality and as a result of intertwined dimensions [32,33]. Previous studies usually concentrated on source credibility and stressed the significant influence of source credibility perception on retweeting behavior in sports news [34] and health information diffusion [35], the importance of content credibility was overlooked to some extent. In brief, content with high credibility should be internally consistent and clearly presented. Many scholars have elucidated the connotation of content credibility, including complete, in-depth, precise, reliable, accurate, unbiased, objective, factual, and fair [36–38]. In a systematic literature review, Sbaffi and Rowley [39] listed dozens of content features influencing trust judgments and credibility perception, which means credibility is a typical compound concept made up of various subdimensions [40]. As a result, it is impossible to investigate all elements of credibility in one study. The proper approach is to pick out the critical factor based on the specific research context.

Completeness is a pivotal part of content credibility, especially in online help-seeking during COVID-19. As the name suggests, it means how much information does an individual disclose and the clarity of the provided information [41]. Completeness contributes to the perception of health information quality on the Internet [42]. Firstly, more complete information means more additional information, which increases the "real world feel" [43]. More detailed information implies more communication cues, improves the social presence of communicators in nonverbally environments [44]. Secondly, completeness is directly bound up with the amount and depth of self-disclosure. Altman and Taylor [28] argued that breadth is the amount of disclosed information, while depth is the intimacy of disclosed information. A complete self-disclosure can effectively convey individuals' needs and reveal who they are, also benefits the likelihood of receiving social support [45]. In this study, we extrapolate that online help seekers' willingness to divulge their personal information helps the audience comprehend their identities. Besides, a complete expression of disease development or health condition

usually demonstrates the seriousness of the current problem, improves credibility perception, and eventually leads to successful support provision. We propose the following:

H1: Completeness of individual information in online help-seeking posts is positively associated with retransmission.

H2: Completeness of disease status in online help-seeking posts is positively associated with retransmission.

## Proximity

Proximity can be interpreted as a form of perceived psychological distance bound up with the construal level theory (CLT) proposed by Trope and Liberman [46]. The psychological distance is defined on spatial, temporal, social, and hypothetical dimensions—those distances further influence how people think about an object or event [46,47]. Substantial studies have scrutinized the impacts of proximity on an individual's responses to others' misfortune, and the main mechanisms can be elaborated in two aspects. Firstly, high proximity implies low perceived distance, which increases empathy and compassion toward the victim, while low proximity weakens it [48,49]. Secondly, proximity links to the perceived trustworthiness of specific information or source. Generally speaking, people often conceive a proximal source as more credible than a distant one [4,35]. Driven by those two threads, when engaged in online activities, high proximity promotes information sharing [50] and information adoption [51]. The work of Huang et al. [30] also manifests the evident influences of physical and emotional proximity on online information seeking under the crisis context.

　　Based on the above rationale, in the supportive communication context, high proximity induces compassion and sympathy more efficiently, which has a potential impact on information diffusion. Likewise, some studies noted that reciprocal social support behaviors frequently occur in close social proximity circumstances [50] and a higher social presence environment arousing psychological proximity [44]. Thus, we propose the following hypothesis:

H3: High proximity triggers more retransmission than low proximity.

## Support typology

People give support to others in multiple ways, while how different types of social support are associated with the seeking-provision remains underexplored. A considerable number of studies summarized distinct social support types and put forward some conceptual frameworks [52,53]. Some scholars conceptually divided social support into four categories, including emotional support (e.g., expression of empathy, love, trust, and caring), informational support (e.g., advice-giving, providing suggestions, sharing information), appraisal support (i.e., offering information for self-evaluation), and instrumental support (i.e., providing tangible support) [52–54]. Others extended the categories of social support by incorporating emotion, esteem, information, network, and tangible assistance [55–57]. Moreover, the social support inventory of UCLA (the University of California at Los Angeles) proposes a classification framework consists of three types: information or advice, tangible assistance or aid, and emotional support [58].

　　For concision and clarity, also following the previous operation [59–61], we merged the existing frameworks into two major types: emotional support and instrumental support. The former one typically refers to needs regarding caring, empathy, love, and trust [59–62], while the latter one denotes the requirements of instrumental resources and practical help [53,59]. The interplay between social support types and support provision is still under-investigated,

particularly the comparison of different types' outcomes has not been thoroughly examined in the online context. This study intends to remedy the defects by discussing how distinct types of social support induce different feedbacks. Therefore, we raise a research question:

RQ1: What is the relationship between support seeking types and retransmission?

## Disease severity

A patient's demands vary by the disease phase because each phase presents a new set of challenges and concomitant opportunities [63,64]. COVID-19 is a relatively novel virus with severe clinical manifestation, and it even incurs death [65]. However, bounded by the shortage of testing capacity in COVID-19's early stage, not all infected patients can get diagnosed formally on time [4]. As a result, suspected cases and confirmed cases were treated differently, receiving different medical services. Before the establishment of makeshift hospitals, patients who have mild symptoms but not get laboratory-confirmed often isolated themselves at home, while those who got confirmed were admitted to hospitals in a relatively short time. Thus, it is reasonable to infer that help seekers in different disease phases would get dissimilar attention. And normally, the more serious the illness, the more urgent the help-seeking and the more likely for the patient to obtain support provision. Accordingly, we posit:

H4: Disease severity expressed in online help-seeking posts is positively associated with retransmission.

## Emotion

Previous studies have proven different emotions affect social decisions differently [66,67]. The logical cues and emotional cues always work in parallel, and emotions sometimes surpass factual information and deliberation [10]. Recent studies regarding emotions and preventive practices (e.g., wear a face covering) during the COVID-19 period indicated that negative emotions mediate the relationship between gender difference and face covering usage [68]. Since rely on reasoning and rely on emotions often occur simultaneously [69], we decide to dive deep into negative emotions in our study for positive feelings are rarely exist in help-seeking posts. Among a plethora of emotional types concerning illness [70], fear is the most prominent emotion during pandemic times, which is highly contagious and makes people feel imminent threats easily [10,71]. Lazarus [72] and Dillard et al. [73] summarized primary discrete emotional kinds in health communication. Apart from fear, anger and sadness are the other two dominating emotions. Anger always connected with removing obstacles, while sadness denotes a manifestation of a sense of failure [72–74]. However, it is still not clear how distinct emotional types induce different retweeting effects. Therefore, we formulate the following research question:

RQ2: What is the relationship between emotional types and retransmission?

## Materials and methods

### Data collection

We choose China as our research field for the following reasons. Firstly, Wuhan, the capital of Hubei province, is the earliest epicenter of the COVID-19 outbreak [75]. However, the Chinese government adopted a series of prevention and control measures to bring the virus under control, almost tamed the spread of COVID-19 in mainland China [76]. China's experience can be an essential reference to other countries, especially in current global pandemic time.

Secondly, with the rapid development of Internet facilities in China, Internet penetration has reached 61.2% in China until June 2019, along with a continuous growing scale of social media users [77]. Some social media platforms play the role of organizing campaigns, promoting the formation of civil society and the public sphere in China [78]. In today's crisis period, Chinese social media exerts the potential to mobilize collective intelligence to overcome difficulties, inspiring other countries to utilize social media to solve health-related problems and cross tough barriers.

Sina Weibo, one of the most popular social media platforms in China, was selected as the sample pool. According to Weibo's user report in 2018, this social media service has accumulated more than 0.4 billion monthly active users and nearly 0.2 billion daily active users until Q4 of 2018 [79]. Weibo has been proved as a vibrant discussion platform and help-seeking space during major social events, especially in the COVID-19 period [4,5]. All authors went through relevant help-seeking posts on Weibo and picked up pertinent keywords. After screening and merging, three pairs of keyword combinations were determined as search terms, including "pneumonia + ask for help" ("肺炎 + 求救" in Chinese), "pneumonia + seek help" ("肺炎 + 求助" in Chinese), and "pneumonia + help me" ("肺炎 + 救救" in Chinese). Date range starts from Jan. 20th, 2020, which is the date that Nanshan Zhong confirmed human-to-human transmission [80], and ends at Mar. 1st, 2020, as the closing date of the first Fang Cang makeshift hospital in the epicenter Wuhan City, indicating the epidemic had been roughly under control in China [81]. We retrieved and collected the data on June 3, 2020.

For now, Sina Weibo doesn't prohibit automated data access explicitly. Previous studies have adopted web crawling scripts to collect Sina Weibo posts related to COVID-19 [4,82]. Besides, our study has been reviewed and approved by the Ethics Review Committee of Tsinghua University, China (No. THU202023). Thus, we took a similar strategy to write a web crawler in Python programming language to retrieve all qualified posts in the designated date range. 34,088 posts were collected in the first round, and 9,826 posted remained after removing duplicated and unqualified posts. We conducted a random sampling to extract a representative sample from the overall corpus because of the large amount of data. Following the American Association for Public Opinion Research's (AAPOR) suggestion [83], we set a 95% confidence level and a 3.5% sampling error, as typical thresholds in social science research. It turns out that the minimum sample size should be 726. Thus, we randomly sampled 727 posts for further analysis. Every single post acts as the analysis unit in our study, contains the user ID, user name, user's registration time, post time, number of followers, number of retweets, content, images' links, authentication type, post's link.

## Measurement

The measurement scheme was built on the literature review. Emotion type is composed of fear, anger, sadness, and others. Support typology contains emotional support seeking, instrumental support seeking, and no specific kind of support seeking. Proximity was operationalized into reporting self-illness, reporting others' illness, and both. Disease severity consists of three kinds: suspected case, confirmed case, and others. Completeness is further decomposed into the completeness of individual information and disease status. Detailed meanings of those categories are listed in Table 1. Number of followers, posting frequency and some other indicators are incorporated into analysis as control variables according to existing research [15,84].

## Coding

A pilot study with 100 posts was conducted to test the coding scheme and train the coders. The pilot study validated the effectiveness of the proposed coding scheme and the accuracy of

the corresponding classifications. Three coders major in communication studies were recruited to code the posts for inter-coder reliability evaluation. The average Krippendorff's alpha coefficient was 0.753 in the first round, which is slightly below the acceptable level. We then retrained the coders, resolved all discrepancies, and sampled another 114 posts from the corpus randomly for a new round trial coding. The average reliability coefficient improved to 0.873 in the second round, which means highly consistent among the coders. All coders then performed coding work on the remaining posts independently.

## Analysis

Since the dependent variable is the number of retweets, traditional linear regression models are inadequate for modeling this kind of highly skewed count variable. Four count models: Poisson regression, negative binomial regression, zero-inflated Poisson regression, and zero-inflated negative binomial regression were compared to fit the data. Firstly, the conditional variance of the outcome variable far exceeds the conditional mean on most categorical explanatory variables, which violates the underlying assumption of Poisson regression. Secondly, we compared the negative binomial regression with the zero-inflated negative binomial regression. Fig 1 displays the residuals from the two tested models, and small residual distribution indicates the good-fitting of the corresponding model.

Besides, the BIC indicator and the sum of Pearson also approved the conclusion that the negative binomial regression model (BIC = 243.908, Sum of Pearson = 7.887) is more preferred to the other one (BIC = 265.635, Sum of Pearson = 27.138). Table 2 shows the descriptive statistical results of all variables.

## Results

Table 3 summarized the results of negative binomial regression models. Model A contains all control variables, while Model B contains both control variables and principal independent variables. All coefficients are evaluated based on robust standard errors.

With regard to the completeness, for each one-unit increase in disclosing individual information, the expected log count of the number of retweets increased by 0.565 ($p < 0.001$, IRR = 1.759). However, the correlation between completeness of disease status and the number of retweets is insignificant. Thus, H1 was supported but H2 was rejected.

When it comes to proximity, reporting self-illness (B = 1.853, $p < 0.05$, IRR = 6.377) and both self and others' illnesses (B = 1.634, $p < 0.05$, IRR = 5.126) are expected to receive more retweets than only reporting others' illnesses. H3 was supported.

For support seeking type, the expected log retweet count of instrumental support seeking is significantly more than emotional support seeking (B = 2.330, $p < 0.001$, IRR = 10.279), no specific support seeking comes afterwards (B = 1.400, $p < 0.01$, IRR = 4.054). RQ1 was answered.

H4 discusses the relationship between disease severity and retransmission. Compared with other illness stages, posts about suspected cases receive more attention (B = 1.686, $p < 0.001$, IRR = 5.398), confirmed cases afterwards (B = 1.583, $p < 0.01$, IRR = 4.868). H4 was rejected.

RQ2 asks how emotional types affect help-seeking diffusion. The expected log retweet count for fear is lower than the expected log count for anger (B = -2.725, $p < 0.001$, IRR = 0.066), followed by sadness (B = -2.749, $p < 0.001$, IRR = 0.064).

Borrowing former scholars' experience [11], we further conducted a sensitivity test using the MANOVA analysis for cross-validation. The robustness check results (both coefficient size and significance) are close to the negative binomial regression, proving the accuracy of our estimation.

**Table 1. Concepts and corresponding variables.**

| Concept | Variable | Category | Connotation | Example[a] |
|---|---|---|---|---|
| Completeness | Completeness | Completeness of individual information | Disclosure of demographic and biographic information, including name, age, ID number, phone number, geographical information, photo [28,85]. | Name: Real name of the patient, such as Aixia Li<br>Age: Such as 65<br>ID number: ID card number (18 digits in China)<br>Phone number: Phone number of the patient or related person<br>Geographical information: Such as Qibiao Community, Hongshan Ave.<br>Photo: Photo of the patient |
| | | Completeness of disease status | Disclosure of disease status, including disease development, diagnostic report, chest X-ray photo, medical record, underlying disease, personal experience [45,86]. | Disease development: "My father couldn't get out of bed four days ago. In the next two days, he became unconscious!"<br>Diagnostic report: Such as blood test report<br>Chest X-ray photo: Photo of chest X-ray<br>Medical record: Such as treatment record<br>Underlying disease: Such as history of hypertension<br>Personal experience: "I have asked the @Wuhan Hotline account and Guanshan Community for help, but none of them can arrange hospitalization." |
| Proximity | Reporting type | Reporting self-illness | Seeking help by disclosing one's own and immediate family members' COVID-19 sickness [4]. | "I was infected with coronavirus, need help." |
| | | Reporting others' illness | Seeking help by mentioning or reporting others' COVID-19 sickness [4]. | "I forward this post from a confirmed patient in Wuhan City, he was infected with COVID-19 but is in a severe shortage of medical care." |
| | | Both | Reporting self-illness and others' illness simultaneously. | "I got infected when taking care of my sick mother. My friend, who lives outside Hubei Province, also infected with this pneumonia." |
| Support typology | Support seeking type | Emotional support seeking | Emotional support involves acting as a confidant for someone, providing empathy or other positive affection toward people who suffer from misfortune [59–61]. | "I have a low fever these nights, so afraid. Can someone give me some comfort?" |
| | | Instrumental support seeking | Instrumental support means offering assistance tangibly or physically, such as donating money, providing medical supplies to someone in sickness [59–61]. | "My mother is in urgent need of a sickbed, can someone help me contact an available hospital?" |
| | | No specific kind of support | Not mention any specific kind of support seeking. | "Who can help them?" |
| Severity | Stage of illness | Suspected | Patients with some COVID-19 infection symptoms but not get laboratory-confirmed [87]. | "I took a chest X-ray, and the doctor said I am a suspected case. But I haven't taken a nucleic acid test yet." |
| | | Confirmed | Patients who get laboratory-confirmed [87]. | "The nucleic test is positive. It turns out I was infected with the coronavirus." |
| | | Others | Unknown status. | Illness status is not mentioned. |
| Emotion | Emotional type | Fear | An emotion experienced in anticipation of some specific pain or danger. Fear is predicated on the belief that the individual faces impending danger over which he or she may have little or no control [73]. | "At present, Mr. Zhao is in critical condition. He has been sent to the intensive care unit of Central South Hospital. He is in urgent need of blood plasma of type O from recovered patients." |
| | | Anger | The health system and government may evoke anger due to their incapability to provide the necessary protection. Particular individuals can also stir up anger for transmitting the disease [88]. | "I'm furious at the work efficiency of local hospitals!!!" |
| | | Sadness | Sorrow and sadness are commonly experienced for loss [88] and irrevocable failure to meet the goal [73]. | "I feel too sad and helpless in front of this pneumonia, will you help me?" |
| | | Others | Other emotions except fear, anger, and sadness. | "I hope everything will get better soon." |

[a] All examples come from posts in our research corpus (translated from Chinese).

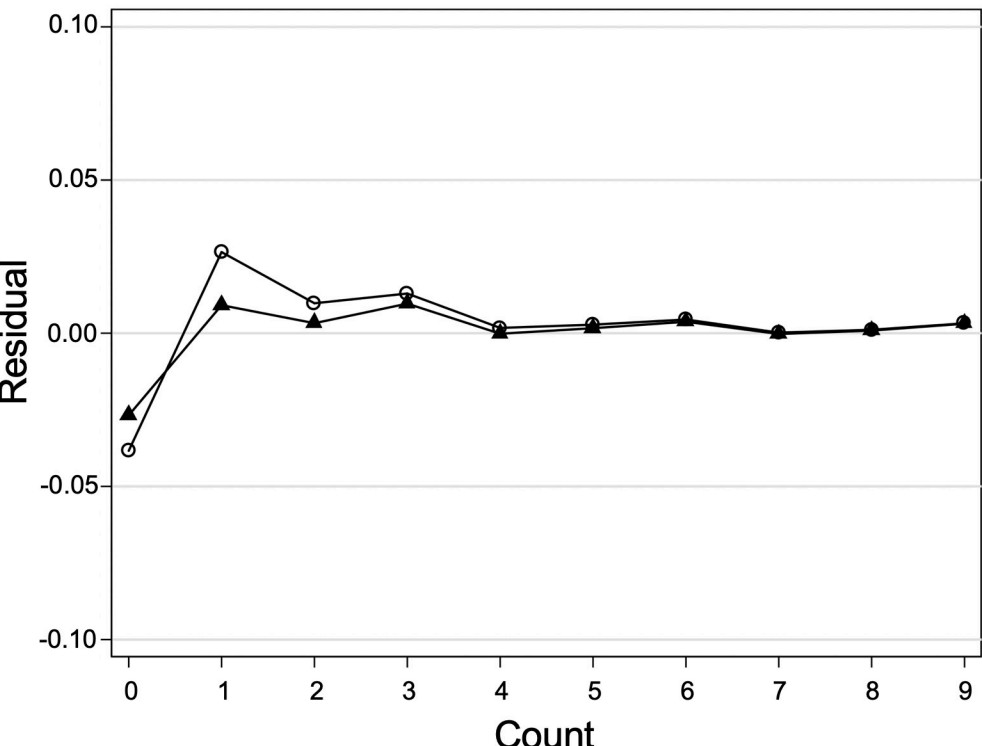

**Fig 1. Model comparison based on residuals.** The line with hollow circles represents the zero-inflated negative binomial regression model, and the line with solid triangles represents the negative binomial regression model. Small residuals are indicative of good-fitting model, the model with the line closest to zero should be considered for our research data.

## Discussion

First of all, the findings of the completeness part are in concert with one recent research. Pan et al. [85] found that support-seeking posts with peripheral self-disclosure elicit lower perceived anonymity in the support-provider side, thus increasing the trustworthiness of the support-seeking messages and improving the quality of advice. Peripheral self-disclosure denotes biographical data, including name, age, gender, geographical information [85], which is precisely the "completeness of individual information" defined in our study. From the perspective of uncertainty reduction theory, message without clear source identity is more likely to be perceived as low credibility, further impeding support-providers' engagement in supportive communication [89,90]. In other words, detailed personal information disclosure can effectively diminish perceived anonymity and demonstrate the vulnerability of help-seekers, which is pivotal in text-based online anonymous settings.

On the contrary, the completeness of disease status fails to trigger retweet behavior. One possible explanation could be the high threshold of medical knowledge posed a high demand for laypersons, resulting in an invisible "communication gap" [91]. The diagnostic report, chest X-ray photo, or medical record requires expertise to understand, which seems impossible to most Weibo users. However, this does not mean that the description of the disease is not important. To improve the communication effect, symptoms and development of the illness should be described in detail through simple words to make it easily understood by ordinary people. This rule could be verified by the positive correlation between the text length and the retransmission [92].

**Table 2. Descriptive statistical results of all variables.**

| Variable | N | Mean (SD) | Min, Max | Description |
|---|---|---|---|---|
| Retransmission | 727 | 1029.779 (9506.688) | 0, 160129 | Number of retransmissions |
| Authentication | 727 | Binary | 0, 1 | 1 being authenticated as influencer (17.33%) |
| With image | 727 | Binary | 0, 1 | 1 being post with image (34.25%) |
| With link | 727 | Binary | 0, 1 | 1 being post with extra link (5.91%) |
| With hashtag | 727 | Binary | 0, 1 | 1 being post with hashtag (74.69%) |
| Mention other user | 727 | Binary | 0, 1 | 1 being post mentions other user (14.31%) |
| Followers (logarithmized) | 727 | 2.498 (1.261) | 0, 7.098 | Number of followers |
| Posting frequency | 727 | 1.839 (7.223) | 0.001, 123.282 | Average number of posts sent by the user per day |
| Text length (logarithmized) | 727 | 2.137 (0.442) | 1.041, 3.217 | Length of post |
| Growth rate of confirmed cases | 727 | 0.147 (0.576) | -0.664, 6.520 | Daily growth rate of national confirmed cases |
| Emotion type | 727 | Four categories | 1, 4 | 1 being fear (42.78%); 2 being anger (11.00%); 3 being sadness (33.70%); 4 being others (12.52%) |
| Support seeking type | 727 | Three categories | 1, 3 | 1 being instrumental support seeking (53.65%); 2 being emotional support seeking (13.76%); 3 being no specific kind of support seeking (32.60%) |
| Proximity | 727 | Three categories | 1, 3 | 1 being reporting self-illness (7.43%); 2 being reporting others' illness (89.13%); 3 being both (3.44%) |
| Disease severity | 727 | Three categories | 1, 3 | 1 being suspected (28.47%); 2 being confirmed (32.46%); 3 being others (39.06%) |
| Completeness of individual information | 727 | 1.915 (1.771) | 0, 6 | Level of detail of demographic and biographic information. 0 = disclose none of the corresponding components, 6 = disclose all corresponding components. |
| Completeness of disease status | 727 | 1.505 (1.640) | 0, 6 | Level of detail of disease status. 0 = disclose none of the corresponding components, 6 = disclose all corresponding components. |

The impact of proximity on information transmission is consistent with the previous studies [30,35], which demonstrates that a high level of proximity triggers more retransmission. Proximity was operationalized into reporting self-illness, reporting others' illness, and reporting both in our study. On the one hand, proximity can be interpreted as a type of constructed imaginary relationship with "others," or a perceived connection between individuals [48]. Imaginary relationship plays a vital role in eliciting emotional or informational responses to suffering. Reporting self-illness, reporting others' illness, and reporting both represent different levels of perceived psychological distance, lead to varying amounts of information retransmission. On the other hand, the rhetoric school emphasizes the effect of "narrative distance" on audiences' emotional involvement [93]. Specifically, narrative perspectives (first-person versus third-person) can influence victim blame and supporting intention by affecting the perceived psychological distance [94]. Follow this thread, reporting self-illness and reporting others' illness can be separately associated with first-person and third-person narrative perspectives, stimulating disparate psychological distances and leads to different responses toward the patients eventually. In other words, reporting self-illness usually creates a sense of telling firsthand experience, which fosters a perception of reliability and authenticity on the support-provider side. It is especially true in the health communication context because a direct expression always outperforms interpreting others' stories. Thus, it is critical to add information about "myself" in a help-seeking message if one intends to gain extensive attention.

Although social network services generally offer substantial opportunities for social support transactions, their potential to provide emotional and instrumental support may differ [95]. In

**Table 3. Results of negative binomial regression models.**

| | Model A: Retransmission (n = 727) | | Model B: Retransmission (n = 727) | |
|---|---|---|---|---|
| **Factors** | **B (95% CI)** | **P-value** | **B (95% CI)** | **P-value** |
| **Constant** | -7.980 (-10.280–5.680) | <0.001 | -6.810 (-9.165–4.455) | <0.001 |
| **Control variables** | | | | |
| Authentication | -0.628 (-1.652–0.395) | 0.229 | -1.352 (-2.100–0.603) | <0.001 |
| Followers (logarithmized) | 1.946 (1.474–2.419) | <0.001 | 1.809 (1.515–2.103) | <0.001 |
| Posting frequency | -0.159 (-0.190–0.128) | <0.001 | -0.116 (-0.146–0.086) | <0.001 |
| Image *vs.* no image | 1.839 (1.174–2.505) | <0.001 | 0.977 (0.244–1.710) | 0.009 |
| Link *vs.* no link | 2.219 (0.467–3.971) | 0.013 | 2.692 (1.103–4.282) | 0.001 |
| Hashtag *vs.* no hashtag | 0.492 (-0.288–1.273) | 0.216 | 0.651 (-0.046–1.348) | 0.067 |
| Mention other users *vs.* not mention | -0.421 (-1.422–0.581) | 0.410 | 0.842 (-0.120–1.805) | 0.086 |
| Text length (logarithmized) | 3.077 (2.202–3.951) | <0.001 | 1.465 (0.519–2.412) | 0.002 |
| Growth rate of confirmed cases | -0.065 (-0.284–0.154) | 0.562 | 0.018 (-0.189–0.224) | 0.867 |
| **Independent variables** | | | | |
| Completeness of individual information | | | 0.565 (0.343–0.787) | <0.001 |
| Completeness of disease status | | | 0.046 (-0.274–0.366) | 0.779 |
| Reporting self-illness *vs.* reporting others' illness | | | 1.853 (0.397–3.309) | 0.013 |
| Reporting both *vs.* reporting others' illness | | | 1.634 (0.099–3.169) | 0.037 |
| Instrumental support *vs.* emotional support | | | 2.330 (1.415–3.245) | <0.001 |
| No specific support *vs.* emotional support | | | 1.400 (0.449–2.350) | 0.004 |
| Suspected *vs.* other stage of illness | | | 1.686 (0.749–2.623) | <0.001 |
| Confirmed *vs.* other stage of illness | | | 1.583 (0.606–2.559) | 0.001 |
| Fear *vs.* anger | | | -2.725 (-3.824–1.626) | <0.001 |
| Sadness *vs.* anger | | | -2.749 (-3.863–1.634) | <0.001 |
| Other emotion *vs.* anger | | | -2.184 (-3.574–0.794) | 0.002 |
| **Model fit** | | | | |
| Log pseudolikelihood | -2525.923 | | -2444.551 | |
| Wald $\chi^2$ | 298.530 | | 666.600 | |
| Pseudo $R^2$ | 0.065 | | 0.095 | |
| AIC | 5073.847 | | 4933.101 | |
| BIC | 5124.325 | | 5034.058 | |

this study, more than half of posts contain instrumental support seeking intention, far exceed emotional support seeking and aimless support seeking. Instrumental support seeking posts receive more retransmission. This result suggests that in the health communication field, especially during a severe pandemic, it is inevitable to face the explosive growth of information. Under this circumstance, attention becomes a scarce but valuable resource [96]. Compared with nihilistic emotional support seeking, instrumental support seeking focused on improving one's health condition, demonstrating direct material needs, and showing the willingness to receive immediate help. Besides, based on Vitak and Ellison's work [97], many people are reluctant to express emotional needs online because they do not want to appear "needy." Emotional support also articulates with intimacy, a prerequisite for emotional communication between interactive partners [98]. The anonymity feature of online communication hinders help seekers' desire for emotional support, impedes the occurrence of in-depth emotional connection [99]. Additionally, the dissemination effect of aimless support seeking also surpasses the emotional one. One possible reason is that there exist many similar posts with intense emotions (e.g., fear toward COVID-19). The audience may immune to those contents after long-lasting exposure, thus mitigate the desire to forward such posts.

Regarding severity, existing studies revealed that giving priority to the worst-case when allocating health resources in pandemics is a fundamental principle [100–102]. However, our findings go against this principle. One significant reason is that demands in posts about suspected cases are easier to meet than posts about confirmed cases. Suspected patients are eager for testing kits and opportunities to visit doctors, while confirmed patients are craving for sickbeds, plasma, and even transfer to another hospital. The former type is more achievable, along with less probability of failure. Impelled by difficulty comparison, retweeters incline to disseminate help-seeking messages which are easy to implement.

When it comes to emotion, posts expressing anger received more retransmission than fear, sadness, and other kinds of emotions. Based on our observation, anger mostly stems from hospitals' unfair treatment or official institutions' delayed responses. It is reasonable for social media users to stand on the "just side" to support the unfortunate patients. By retweeting the angry posts, retweeters vented their feelings, intended to get more attention, and urged relevant departments to take effective measures. In our research corpus, sadness is the second common emotion. However, it received the least retransmission. This can be attributed to sadness's intrinsic characteristics: posts with sadness mainly express disappointment toward reality but resist to advocate practical attempts to change reality. This finding implies that when seeking help online, one should stress the principal problem and avoid simple catharsis. Similarly, ingeniously using prevailing emotion contributes to receiving sympathy and pragmatic feedback from others.

## Conclusion

COVID-19 poses enormous threats to the whole world. Due to the unbalanced distribution of medical resources and inadequate response at the early breakout stage, many patients (both suspected and confirmed), along with their relatives or friends, turned to social media to seek support. This study explores the driving forces behind online help-seeking post transmission in a global pandemic period by emphasizing the content characteristics. Completeness, proximity, support typology, disease severity, and emotion composed the analytical framework. By employing content analysis, 727 randomly sampled help-seeking posts were analyzed based on a coding scheme we construct. Negative binomial regression reveals that posts release anger, express instrumental support seeking intention, report self-illness, expound suspected cases' conditions, and have detailed individual information disclosure are likely to have more retransmission.

Coronavirus is still spreading fast around the world. As one of the first countries to restrain the spread of the epidemic, China provides a valuable experience to other countries stuck in a dilemma. As the first online help-seeking analysis research in the COVID-19 period, this study offers some insights about health communication via social media, especially how to develop a potent help-seeking post in public health emergencies. However, a couple of limitations need to be mentioned. First, retransmission of post, as the core dependent variable in our study, is a multidimensional concept. For example, Wang et al. [11] decomposed retransmission into "scale" and "structural virality." Limited by research resources and time, we failed to elucidate information diffusion comprehensively. Second, although China's experience is representative, whether this pattern can be generalized into other contexts worth carefully examining. Future studies should conduct more explorations on other social media platforms (e.g., Twitter) to verify the reliability and validity of our results and summarize effective online help-seeking strategies in diversified environments.

## Acknowledgments

The authors want to thank Prof. Jing Qian (Department of Psychology, Tsinghua University) for her tremendous help in the ethical review process.

## Author Contributions

**Conceptualization:** Chen Luo, Yuru Li, Anfan Chen, Yulong Tang.

**Data curation:** Anfan Chen.

**Formal analysis:** Chen Luo, Yuru Li, Anfan Chen.

**Methodology:** Chen Luo, Yuru Li, Anfan Chen.

**Project administration:** Chen Luo.

**Visualization:** Chen Luo.

**Writing – original draft:** Chen Luo, Yuru Li, Anfan Chen, Yulong Tang.

**Writing – review & editing:** Chen Luo, Anfan Chen.

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
