## [Decision Letter · Decision Letter 0]

16 Jul 2020

PONE-D-20-18830

What triggers online help-seeking retransmission during the COVID-19 period? Empirical evidence from Chinese social media

PLOS ONE

Dear Dr. Li,

Thank you for submitting your manuscript to PLOS ONE. After careful consideration, we feel that it has merit but does not fully meet PLOS ONE’s publication criteria as it currently stands. Therefore, we invite you to submit a revised version of the manuscript that addresses the points raised during the review process.

Please find below the reviewer's and mine's comments.

We look forward to receiving your revised manuscript.

Kind regards,

Valerio Capraro

Academic Editor

PLOS ONE

Journal Requirements:

2. In your Methods section, please include additional information about your dataset and ensure that you have included a statement specifying whether the collection method complied with the terms and conditions for the website.

Additional Editor Comments (if provided):

I have now collected one review from one expert in the field. The reviewer is positive but suggests several revisions. Therefore, I would like to invite you to revise your work along the reviewer's comments. Moreover, I would like to add a couple more comments that I have had while reading the manuscript. (i) in the general introduction, the "perspective article" about what social and behavioural science can do to support pandemic response, published by Van Bavel et al. in Nature Human Behaviour, can be a useful general reference. (ii) you correctly say that retweeting can be seen as a form of prosocial behaviour; at the same time, many of these posts are very emotional. Therefore, it seems that your research question is related to the work on how emotions affect prosocial behaviour. There is a huge literature about this (see Capraro 2019) for a review. And recently emotions have been also linked to mask wearing (Capraro & Barcelo 2020a, 2020b). I think this potential link should be discussed. Of course it is not a requirement to cite these works, but I am mentioning them because they look very related.

I am looking forward for the revision.

References

Capraro, V. (2019). The dual-process approach to human sociality: A review. Available at SSRN 3409146.

Capraro, V., & Barcelo, H. (2020a). The effect of messaging and gender on intentions to wear a face covering to slow down COVID-19 transmission. arXiv preprint arXiv:2005.05467.

Capraro, V., & Barcelo, H. (2020b). Priming reasoning increases intentions to wear a face covering to slow down COVID-19 transmission. arXiv preprint arXiv:2006.11273.

Van Bavel, J. J., et al. (2020). Using social and behavioural science to support COVID-19 pandemic response. Nature Human Behaviour, 1-12.

Reviewers' comments:

Reviewer's Responses to Questions

**Comments to the Author**

1. Is the manuscript technically sound, and do the data support the conclusions?

Reviewer #1: No

2. Has the statistical analysis been performed appropriately and rigorously? 

Reviewer #1: Yes

3. Have the authors made all data underlying the findings in their manuscript fully available?

Reviewer #1: Yes

4. Is the manuscript presented in an intelligible fashion and written in standard English?

Reviewer #1: Yes

5. Review Comments to the Author

Reviewer #1: This work presents results pertaining the factors behind the retransmission/retweeting of online help-seeking posts that emphasized on content characteristics which are information completeness, proximity, support seeking type, disease severity, and emotion related to COVID-19.

This is an in principle interesting topic, however, the current manuscript has numerous concerns including some crucial conceptual and methodological issues that can affect the effective strength of findings.

Major Points

1. How do you conclude to get the 727 posts? Please state sample size calculation.

2. How did you make sure that pneumonia as keywords are related to covid-19?

3. Could the author give one example how variables are extracted from the post.

Minor Points

1. Please elaborate "sender identities are very close" page 6 line 22.

2. The impacts of proximity are reflected in two aspects. Page 9 line 11. Which aspects are you referring to?

3. Please state the necessary attributes in Page 15 line 2

4. What is level of detail (0-6) represent in completeness attribute.

6. PLOS authors have the option to publish the peer review history of their article (what does this mean?). If published, this will include your full peer review and any attached files.

Reviewer #1: No

---

## [Author Response · Author response to Decision Letter 0]

25 Sep 2020

Dear editor and reviewer, 

Thank you very much for providing us an opportunity to revise and respond to reviewers’ valuable comments. Following all these comments, we have thoroughly revised our manuscript. The main changes are summarized as follows:

1. We carefully adjusted the format of this article to meet the style requirements. Also, we added approval from the Ethics Review Committee to justify our research.

2. According to the comments, we significantly expanded our literature review on the relationship between pandemic response and behavioral science (as reflected on Page 3), as well as how emotions affect prosocial behavior (as reflected on Page 10). 

3. We followed reviewers' suggestions to provide more information on conceptual and methodological issues (as reflected on Page 11-12). Moreover, we rechecked our dataset to make everything accurate. Thus, we replaced a few duplicated cases with newly added posts and reevaluated all coefficients in our paper. The final dataset (which has been uploaded to the figshare platform) and statistical results in the current version are entirely accurate and reliable.

4. We refined our statistical reporting with more details to meet the standard of PLOS ONE, including replaced the regression coefficient plot with a more common regression table (see Table 3 on Page 20-21).

 We believe the manuscript is much stronger and technically sound. In the following pages, please find a detailed point-to-point response to the editorial and reviewer comments.

Along with my coauthors, I thank you sincerely for the consideration of publishing our paper on PLOS ONE.

Kind regards,

Yuru Li, Ph.D. (E-mail: yuru.li@foxmail.com)

● Response to Editor

Thanks for your comment, we have revised our format in strict accordance with PLOS ONE's style requirements.

2. In your Methods section, please include additional information about your dataset and ensure that you have included a statement specifying whether the collection method complied with the terms and conditions for the website.

We appreciate this comment, as this issue of terms of service violation is a formidable hindrance to researchers of infoveillance and the Internet in general. First of all, we have included additional information about our data gathering in the Methods section, which incorporates the approval from the Ethics Review Committee (as shown on Page 12, Line 1-4). Then, we would like to further enumerate several reasons to verify our data collection method's reasonability and morality.

1) We collected our research corpus with our self-developed web crawling program from the Sina Weibo platform. The design principle of Sina Weibo is to make user-generated content as visible as possible, which means anyone can read content posted by Weibo users. From this point, we didn't infringe on Weibo users' privacy. Besides, we choose not to disclose the original help-seeking posts, but anyone can see the same content on the Weibo platform using the designated searching criteria in our paper.

2) Two conventional approaches to collecting online data are calling the application program interface (API) and applying self-developed web crawling scripts. As a matter of fact, Sina Weibo has issued a policy announcing that API services are paid services and merely opened to commercial entities [1]. For researchers who are interested in social media, only limited access is provided. If we adopt the official APIs, we can only obtain incompleted data and bear the loss of missing attributes.

3) Many websites (including Weibo) now prohibit, or at least not explicitly permit the use of automated crawling software to access its content. Fortunately, a precedent has been set recently in Sandvig v. Sessions [2], in which Judge John Bates ruled in favor of all researchers to get publicly accessible information on websites under First Amendment protection, despite the possible violation of terms of service. Since Weibo is a publicly-traded company in the U.S., this rule applies to Weibo. We firmly believe this ruling is an important step forward for Internet-related research. It protects researchers like us from unwarranted legal liabilities while all we do is contribute to public knowledge and the public good. 

4) Previous social science research on COVID-19 took similar collecting strategies. For instance, Shen et al. [3] crawled all posts sent by users in their Weibo user pool and applied keywords searching to obtain the research corpus. Hu et al. [4] released a large-scale COVID-19 social media dataset based on Weibo, in which 40 million posts from Weibo users were crawled and recorded (with anonymous identities). We believe the data collecting operation in our research is entirely legal, and this strategy benefits us to cope with the COVID-19 epidemic more timely and efficiently.

3. (i) In the general introduction, the "perspective article" about what social and behavioral science can do to support pandemic response, published by Van Bavel et al. in Nature Human Behaviour, can be a useful general reference. 

(ii) You correctly say that retweeting can be seen as a form of prosocial behavior; at the same time, many of these posts are very emotional. Therefore, it seems that your research question is related to the work on how emotions affect prosocial behavior. There is a huge literature about this (see Capraro 2019) for a review. And recently emotions have been also linked to mask-wearing (Capraro & Barcelo 2020a, 2020b). 

Thanks for your recommendation. We appreciate your advice because the literature is thought-provoking and helpful, and we have absorbed them in our research paper. The details are summarized as follow:

1) The "perspective article" published on Nature Human Behavior provides us with a comprehensive framework to understand retweeting behavior under social and behavioral science. We incorporated this literature in our introduction section, as reflected on Page 3, Line 19-22. Besides, the dual-process approach summarized by Capraro (2019) also helps us to organize the following content more organically. Thus, we mentioned this literature in the introduction section, as reflected on Page 4, Line 4-6.

2) Emotion is a vital factor in affecting retweeting behavior. Capraro and Barcelo's works on the relationship between emotion and preventive practices during the COVID-19 pandemic time is quite pertinent to our research. We referred to those works in the "emotion" section on Page 10, Line 4-7. 

● Response to Reviewer #1

1. Major point 1: How do you conclude to get the 727 posts? Please state the sample size calculation.

Thank you for the comment. According to the American Association for Public Opinion Research (AAPOR), sampling error and confidence level jointly determined the sample size in a random sampling operation [5]. Generally, researchers set the confidence level as 95% and the sampling error between 3% and 6% in social science research. In our study, we adopted a 95% confidence level and 3.5% sampling error to calculate the proper sample size. Combined with the population size (n=9,826), it turns out that we need at least 726 posts to achieve the specified error level (calculation website: https://www.surveysystem.com/sscalc.htm). We also explain this briefly in the paper, as reflected on Page 12, Line 5-10.

2. Major point 2: How did you make sure that pneumonia as keywords are related to COVID-19?

Thanks for your comment. Firstly, in the Chinese context, "COVID-19" (“新型冠状肺炎”), "new coronavirus pneumonia" (“新冠病毒肺炎”), and "new pneumonia" (“新型肺炎”) are common appellations for this epidemic. No matter what the name is, it contains the keyword "pneumonia" (“肺炎”). Thus, adopting "pneumonia" as the search term ensures us get the most comprehensive corpus. 

 Secondly, based on our observation (three coders read almost all the crawled posts), most Weibo users tend to use the word "pneumonia" to describe the COVID-19 and its symptoms within the date range we set. 

 Thirdly, during the pandemic period, the Weibo platform opened several specific hashtags and super topics, which contained highly relevant help-seeking posts sent by infected people and their friends or relatives. Those hashtags are like "help-seeking from pneumonia patients" (“肺炎患者求助”), they did not include the words "new coronavirus" (“新冠病毒”), "new type" (“新型”), or "nCoV-19" (“新冠”). Therefore, the "pneumonia" search term also guarantees us to obtain the most accurate corpus in the specific date range. 

 Besides, we applied keyword combinations as search terms instead of using the "pneumonia" word only. This operation helped us to target the help-seeking posts related to the COVID-19 epidemic. For clarity, we add corresponding Chinese translation of keyword combinations in our paper, as reflected on Page 11, Line 17-18.

3. Major point 3: Could the author give one example of how variables are extracted from the post?

Thank you for the comment. We have added a column (“Example”) in Table 1 to elucidate how those variables are extracted from the post. Contents in the newly added column are translated from posts in our corpus. Besides, our answer to minor point 4 also provides some clues on the operationalization process.

4. Minor point 1: Please elaborate "sender identities are very close" page 6 line 22.

Thank you for the comment. Weibo has evolved into a critical virtual platform for Chinese grassroots to express their opinions [6]. During the COVID-19 pandemic period, the public, or the so-called ordinaries, disclosing their symptoms and asking for help actively on Weibo [3,7]. However, there are three kinds of users on the Weibo platform, including ordinary users, organizational users, and celebrities. The latter two are featured with specific visual symbols, and they are more influential than the ordinary user in the virtual world. By mentioning "sender identities are very close," we intended to indicate that help seekers are mainly composed of ordinary users. Organizational users and celebrities usually occupy more resources and seldom asking for help online. To make our meaning more transparent, we slightly adjust the words in the main text, as reflected on Page 5, Line 15-18.

5. Minor point 2: The impacts of proximity are reflected in two aspects. Page 9 line 11. Which aspects are you referring to?

Thanks for the comment. We have to say your comment comes right to the point because the two aspects here seem confusing. Actually, we intend to explain the underlying mechanisms of how proximity works in the mental process by listing the two aspects. For legibility, we reframe this part by using the "Firstly..., secondly..." sentence pattern and removing the weakly related content (as reflected on Page 7-8). By this operation, key concepts (including proximity, psychological distance, and information processing) are more tightly integrated.

6. Minor point 3: Please state the necessary attributes on Page 15 line 2.

Thanks for this comment. We have supplemented the attributes in our paper, as reflected on Page 12, Line 11-12. Besides, for ease of understanding, we provided two screenshots with annotations in this response letter to illustrate those attributes.

(Two illustrations are inserted here, please refer to the "Response to Reviewers.docx" for details)

7. Minor point 4: What is the level of detail (0-6) represent in the completeness attribute. 

Thank you for the comment. As shown in Table 1, the completeness attribute is composed of two dimensions: the completeness of individual information (stresses the demographic and biographic information), and the completeness of disease status (emphasizes the disease status). For operationalization, we further listed several components under each dimension. If one post contains more components, then the corresponding dimension score increases, which means the sender discloses more relevant information. A score of 0 means the sender disclosed none of the components, while a score of 6 means the sender disclosed all the components. 

 We have supplemented the meanings in Table 2 for clarity. In the meantime, we provide an example here to illustrate this further. The following screenshot is the sample post, and we translated it into English.

(One illustration is inserted here, please refer to the "Response to Reviewers.docx" for details)

Translation: 

My parents both infected with COVID-19 and got confirmed. My father has difficulty breathing and feels sick, but he has not been hospitalized up to now!!!

[Name] Liyong Liu

[Age] 60

[City] Wuhan

[Address] Qibiao Community, Guanshan St. 

[Time of illness] Jan. 22nd, 2020 

[Symptoms] Persistent high fever, cough, chest tightness, dyspnea, diarrhea, and poor spirit

My family lives in the Qibiao Community, Guanshan Ave., Hongshan District, Wuhan City. My father was hospitalized twice in January this year because of stomach diseases. Since then, he has been frail, and my mother was infected with COVID-19 while taking care of him. I have asked the @Wuhan Hotline Weibo account and the Guanshan Community for help, but none of them can arrange hospitalization. So I can only make registration and stay on the waiting list. Yesterday, my father went through further consultation. The doctor pointed out that his lung infection is very severe and has to be hospitalized immediately. However, the Guanshan Hospital is short of sickbeds, and we have no choice but to wait for the community arrangement. Please help me. Please save my parents. I beg the government for a sickbed to get my father treated as soon as possible! My parents lived a hard life before. They suffered from natural disasters, got polio at a young age. They also experienced layoffs in their prime period. It is hard for them to come across the COVID-19 epidemic in their retirement. Please help us! [Location: Guanggu District, Wuhan City]

(Below are the pictures.)

This post is coded as below:

(One illustration is inserted here, please refer to the "Response to Reviewers.docx" for details)

 

References

1. weibo.com [Internet]. Application program interfaces for commercial data [cited 2020 Jul 21]. Available from: https://open.weibo.com/wiki/%E5%95%86%E4%B8%9A%E6%95%B0%E6%8D%AEAPI.

2. Williams J. D.C. Court: Accessing public information is not a computer crime [Internet]. Electronic Frontier Foundation; 2018 [cited 2020 Jul 21]. Available from: https://www.eff.org/deeplinks/2018/04/dc-court-accessing-public-information-not-computer-crime.

3. Shen C, Chen A, Luo C, Zhang J, Feng B, Liao W. Using reports of symptoms and diagnoses on social media to predict COVID-19 case counts in mainland China: Observational infoveillance study. J Med Internet Res [Internet]. 2020 May 28 [cited 2020 Jul 17]; 22(5): e19421. Available from: https://www.jmir.org/2020/5/e19421/

4. Hu Y, Huang H, Chen A, Mao X. Weibo-COV: A large-scale COVID-19 social media dataset from Weibo. arXiv: 2005.09174 [Preprint]. 2020 [cited 2020 Jul 21]: [9 p.]. Available from: https://arxiv.org/abs/2005.09174

5. American Association for Public Opinion Research. Margin of sampling error/credibility interval [Internet]. American Association for Public Opinion Research [cited 2020 Jul 17]. Available from: https://www.aapor.org/Education-Resources/Election-Polling-Resources/Margin-of-Sampling-Error-Credibility-Interval.aspx

6. Lu J, Qiu Y. Microblogging and social change in China. Asian Perspect. 2013;37(3): 305-331.

7. Huang C, Xu X, Cai Y, Ge Q, Zeng G, Li X, et al. Mining the characteristics of COVID-19 patients in China: Analysis of social media posts. J Med Internet Res [Internet]. 2020 May 17 [cited 2020 Jul 17]; 22(5): e19087. Available from: https://www.jmir.org/2020/5/e19087/

---

## [Editor Report · Decision Letter 1]

16 Oct 2020

What triggers online help-seeking retransmission during the COVID-19 period? Empirical evidence from Chinese social media

PONE-D-20-18830R1

Dear Dr. Li,

We’re pleased to inform you that your manuscript has been judged scientifically suitable for publication and will be formally accepted for publication once it meets all outstanding technical requirements.

Kind regards,

Valerio Capraro

Academic Editor

PLOS ONE
---

## [Editor Report · Acceptance letter]

22 Oct 2020

PONE-D-20-18830R1 

What triggers online help-seeking retransmission during the COVID-19 period? Empirical evidence from Chinese social media 

Dear Dr. Li:

I'm pleased to inform you that your manuscript has been deemed suitable for publication in PLOS ONE. Congratulations! Your manuscript is now with our production department. 

Kind regards, 

on behalf of

Dr. Valerio Capraro 

Academic Editor

PLOS ONE